# Tree Resin, a Macroergic Source of Energy, a Possible Tool to Lower the Rise in Atmospheric CO$_2$ Levels

Jaroslav Demko and Ján Machava *

Department of Biology and Ecology, Faculty of Education, Catholic University in Ruzomberok, Hrabovska Cesta 1, 034 01 Ruzomberok, Slovakia; jaroslav.demko@ku.sk
* Correspondence: jnmchv9@gmail.com; Tel.: +421-905-756824

**Abstract:** Tree resin is a macroergic component that has not yet been used for energy purposes. The main goal of this work is to determine the energy content of the resin of spruce, pine, and larch and of wood components—pulp and turpentine. The combustion heat of resin from each timber was determined calorimetrically. Approximately 1.0 g of liquid samples was applied in an adiabatic calorimeter. The energy values of the tree resin (>38.0 MJ·kg$^{-1}$) were 2.2 and 2.4 times higher than that of bleached and unbleached cellulose, and the highest value was recorded for turpentine (>39.0 MJ·kg$^{-1}$). Due to the high heating values of the resin, it is necessary to develop approaches to the technological processing of the resin for energy use. The best method of resin tapping is the American method, providing 5 kg of resin ha$^{-1}$ yr$^{-1}$. The tapped resin quantity can be raised by least 3 times by applying a stimulant. Its production cost compared to other feedstocks was the lowest. Tree resin can be applied as a means of mitigating global warming and consequently dampening climate change by reducing the CO$_2$ content in the atmosphere. One tonne of tree resin burned instead of coal spares the atmosphere 5.0 Mt CO$_2$.

**Keywords:** resin; combustion heat; renewable energy source; wood; carbon sequestration; climate change

## 1. Introduction

According to European Union policy (Directive 2009/28/EC), 20% of the overall energy consumption in the EU by 2020 should be provided by renewable sources. In this respect, adequate findings must be obtained for the characterization and identification of specific biomass types [1,2], because forest biomass quality is closely associated with carbon sequestration, lowering its content in the atmosphere [3]. A more detailed study on the energy issue of resin as an important component of tree biomass is therefore necessary.

Resins are related to a family of extractive substances, including waxes, suberin, cutin, glycosides, alkaloids, and others, that protect the plants against unfavorable biotic and abiotic influences [4–9]. Resin can be chemically characterized as a group of aromatic compounds of a terpene nature, formed by two major classes of substances—the first group of compounds is derived from phenylpropane, and the second is from terpene compounds. Resin is a mixture of various substances, of which resin acids, terpene-type hydrocarbon, terpenoid alcohols, waxes, and solutions of relatively light or volatile terpenes are the most important [10–13]. The terpenes are largely monoterpenes that generally comprise about 25% of the total weight of resin. The remainder of the liquid fraction is chiefly sesquiterpenes (0–20%). Thus, the aggregate of all terpenes in the whole resin is in the range 25–45% [14]. It is generally agreed that terpenes are formed by the polymerization of isoprene (C$_5$H$_8$) from hemiterpenes, through monoterpenes, and up to polyterpenes (including rubber and gutta) [15,16].

Tree resin is a complex mixture of volatile and non-volatile terpenes. Mono-(C$_{10}$H$_{16}$) and sesquiterpenes (C$_{15}$H$_{24}$, turpentine; farnesyl diphosphate, FPP) are generally volatile,

giving fluidity to the resin, and they usually give resin its characteristic odor. When only volatile mono- and sesquiterpenes occur, they often are called essential oils. This designation, however, is misleading because these terpenoids are neither essential to plant metabolism nor true oils; "essential" refers to their essence or fragrance, and "oil" refers to their feel. Turpentine constitutes the largest group of secondary products and derives from a 5C compound, IPP (isopentenyl pyrophosphate). Non-volatile diterpenic acids ($C_{20}H_{32}$, rosin), the most frequent and abundant diterpenoid resin compounds occurring in rosin, are derivatives of abietane, pimarane, and isopimarane skeletons [17,18]. Diterpene acids are particularly important in resin. The doubling (dimerization) of the $C_{15}H_{24}$ (FPP) leads to $C_{30}H_{52}$ compounds, i.e., triterpenes. Triterpenes include a wide variety of structurally diverse substances. These terpene fractions are a natural source of valuable materials for chemical industries [19].

Plant resin can be defined primarily as a lipid-soluble mixture of volatile and non-volatile terpenoid and/or phenolic secondary compounds that are usually secreted in specialized structures located either internally or on the surface of the plant, and of potential significance in ecological interactions [10]. Resin is sometimes confused with oleoresin. Oleoresins are comparatively fluid terpenoid resins with a higher ratio of volatile to non-volatile terpenes. Hereinafter, the resin will mostly be referred to as oleoresin.

After the collection, crude resin conversion into gum turpentine and gum rosin is carried out by steam distillation [20]. Subsequently, these by-products are processed for use in the fabrication of diverse industrial products such as feedstock, cleaners, pine oil, fragrances and flavoring compounds, pesticides, solvents and thinners for paints, and pharmaceuticals products [21–23]. Most *Pinus* species yield pinenes ($\alpha$- and/or $\exists$-pinene) as major compounds in their monoterpenic turpentine fraction. Large amounts of pinenes are used in the flavor and fragrance industry [24]. However, due to their strong odor, they cannot be extensively used as additives in flavors or fragrances; pinenes are chemically converted to more valuable products, such as verbenone, a monoterpene that exhibits an ecological role as an anti-aggregating pheromone (tree protection) [25–30]. Besides this, pinenes might also be used in the production of pharmaceuticals, plasticizers, repellents, insecticides, solvents, perfumery, cosmetics, and antiviral and antimicrobial compounds [26–32]. Some investigations have been carried out with the aim of assessing the economic viability of performing resin-tapping operations [21,33–35].

The combustion heat and the calorific value of stump wood, tree crown components, and assimilatory organs are relatively well known [36–38], and fluctuate depending on tree species. For chestnut and pine, the crown fraction generates the highest energetic values, and the wood produces the lowest. The calorific value per tree is lowest for chestnut (17.419 MJ·kg$^{-1}$), intermediate for Short-Rotation Coppice crops (18.185–18.419 MJ·kg$^{-1}$), and highest for maritime pine (19.336 MJ·kg$^{-1}$) [38]. This is caused by the higher lignin and resin contents in coniferous species. These results are consistent with the trends observed by Telmo [39], who reported higher energetic values for different pine species than for native broad-leaved and autochthonous species in northern Portugal. Wood extractives such as resin, waxes, oils, tannins, and other phenolic substances also have much higher heating values than cellulose and hemicelluloses, and they are abundant in the wood of coniferous species [40]. The contents of terpenes and oleoresins significantly affect the energetic values of "lignocellulosic fuels". Furthermore, Howard [41] has calculated the higher heating values of resin as 15,000 to 16,000 btu/lb (34.890–37.216 MJ·kg$^{-1}$), but the variation in the results was large, as the resin samples were semiliquid and rather inhomogeneous. In a follow-up on the inquiries stated, this work examines the properties of tree resin and its potential for use as a renewable energy source.

*Problems*

Resins are macroergic solids (high energy), formed in a manner similar to the essential oils in specific resin canals. Physiological resin is formed by the metabolic activity of trees, and pathological resin is mainly produced as a result of damage to the trees. The

amount and composition of the resin in various types of wood depends largely on the biochemical taxon of individual species, environmental effects [42], geographic origin [43], local conditions [44,45], etc.

Significant differences in the composition of resin were found not only between different coniferous trees, but also between trees of the same species [46]. Differences were found between the physiological resin composition coming from a healthy tree and the pathological resin flowing from the bark of the injured tree (similar designations: a constitutive and induced resin). Pathological resin consists substantially of terpenes and resin acids, it is fat-free, it provides excellent resin after distilling turpentine, and it is also referred to as callus resin [47]. The resin obtained by extraction from harvested wood cannot also be considered as physiological in origin, because the solvents used are not indifferent to the oleoresin or to its constituents, and the resin obtained by extraction is chemically modified. From a chemical point of view, the oleoresin obtained from growing trees is, therefore, more acceptable and represents a more appropriate approach.

The importance of an energy analysis of dendromass components, oleoresin in this case, is important for obtaining information on their living nature and their high content in some woods. Due to their high energy value, these findings play an important role, for example, in selecting the biochemical taxon of trees, which ensures the highest possible production of energy biomass at energy forest plantations [48,49].

Generally, four pine resin-tapping techniques are used worldwide. The most effective is the American method (Figure 1, an improved V-shape) [50], often referred to as the bark streak method [51].

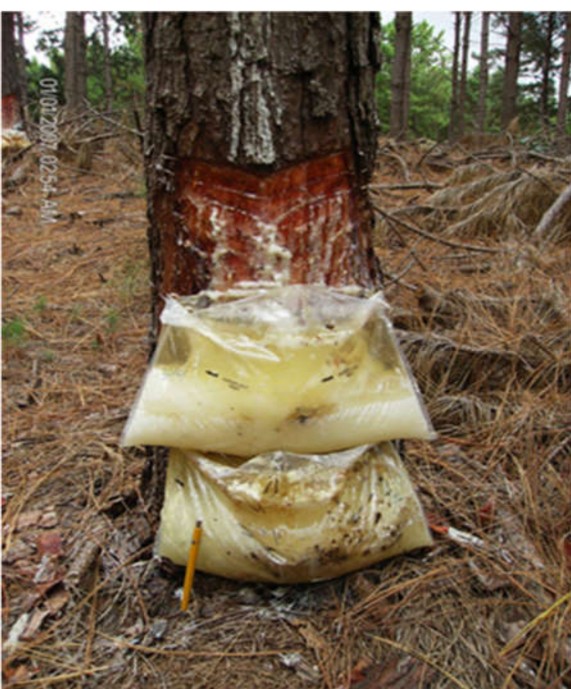

**Figure 1.** American pine resin tapping techniques (V-shaped streaks (2–3 mm wide)). Cut into xylem. Source: [52].

Nowadays, it is well established that tree resin properties depend on key factors, and mainly on climate conditions [53,54], the genetic background, and the environmental effects [42,55]. Additionally, the quality and quantity of resin and its by-products can be influenced by age [56], geographic origin [43], and different stresses, such as low-fertility soils [57], drought [44,58], extreme temperatures [59–61], and the hydrological impact [62]. Further, resin production can be modulated by resin-tapping methods [50], associated with chemical stimulation treatments [63].

Although the history of the tapping and processing of tree resin dates back to the 18th century, energy values, such as the combustion heat and calorific values, are rarely stated for a macroergic substance such as oleoresin. Absent or incomplete findings prompted us to conduct an energy review of the resin of the most economically important conifers in the Central European region: spruce (*Picea abies* (L.) H. Karst), pine (*Pinus sylvestris* L.), and larch (*Larix decidua* Mill.) trees. The main goal of this work is to determine the combustion heats of the oleoresins of these trees, and to compare them with those of further tree wood components. Knowing the energy levels of these substances is important because biomass currently forms the basis of renewable energy sources and forest ecosystem services in Central European conditions.

## 2. Materials and Methods

### 2.1. Sampling Material

The resin samples were taken by a modified streaking method from spruce (*Picea abies* (L.) H. Karst), pine (*Pinus sylvestris* L.), and larch (*Larix decidua* Mill.) trees growing on the Cernova research area (3,3 ha), nearby Ruzomberok (A: 49.101501 N, 19.236523 E, B: 49.100178 N, 19.236632 N, C: 49.100344 N, 19.239303 E, D: 49.101986 N, 19.239519 E), at an altitude of approximately $490 \pm 90$ m asl., and were transported to the laboratory and stored in a freezer prior to energy determination analyses. The tree bark was cut with a groove knife into the sapwood (a modified V-shaped streak), and the resin flowing from there was later collected with attached cups. The resin obtained in this way is not actually physiological resin. However, it is more suitable than that obtained from harvested timber by chemical extraction.

To compare the energy of the resin obtained by the streaking method, a resin component from pulp production, known as turpentine, was used. This is essentially a mixture of resins. The measured energy values of the resin were further compared with the basic constituents of the dendromass, such as bleached and unbleached cellulose and turpentine. These wood constituents were obtained from Mondy SCP, Ltd. (Ružomberok, Slovakia), a pulp and paper complex in Ruzomberok.

### 2.2. Assessment of Energy—Calorimetry

Approximately 1.0 g of liquid samples was weighted on the Denver Instrument SI-234 scale with a precision of 0.0001 g and placed into a 3 mL metal crucible. After weighing, determination of the combustion heat was accomplished in an adiabatic bomb calorimeter (Model IKA Calorimeter C400). The resin sample was completely incinerated at $3.0 \pm 0.3$ MPa in a pure oxygen environment. The heat emitted during incineration was transferred to the calorimeter fluid, whereby the heat capacity for each calorimeter was specified. From the difference in temperature ($\Delta T$) between the initial condition and after incineration, the level of energy released from the sample material as well as the combustion heat was calculated by the respective equation [64]. At least one replicate determination was carried out for each resin sample.

### 2.3. Statistical Evaluation

The data determined were tested for normality using the Kolmogorov-Smirnov test and the SPSS statistical packet. The normal distribution of the combustion heat data of pine and larch was confirmed by the *p*-values of 0.141 and 0.200. In the spruce data, the first two values were outliers, and the data did not meet the normality presumption ($p = 0.026$). No significant differences between the combustion heat values determined for pine, spruce, and larch were indicated by the non-parametric Kruskal-Wallis test ($p < 0.509$). Statistical parameters of combustion heat values for pine (*Pinus sylvestris* L.), spruce (*Picea abies* (L.) H. Karst), and larch (*Larix decidua* Mill.) are stated in Table 1.

**Table 1.** Statistical parameter of combustion heat of tree resin (MJ·kg$^{-1}$).

|  | Pine | Spruce | Larch |
|---|---|---|---|
| Minimum | 36.00 | 34.60 | 36.30 |
| Maximum | 40.10 | 40.00 | 39.70 |
| Range | 4.10 | 5.40 | 3.40 |
| Mean | 38.60 | 38.37 | 38.33 |
| Stand. error | 0.30 | 0.35 | 0.23 |
| Stand. deviation | 1.30 | 1.53 | 0.99 |
| Variance | 1.70 | 2.33 | 0.97 |
| Skewness | −0.65 | −1.51 | −0.57 |
| Kurtosis | 0.52 | 2.05 | 0.52 |
| Number | 19.00 | 19.00 | 19.00 |

## 3. Results

The renewable biomass resources studied in the past for energy purposes comprise many herbaceous and woody plants. In the composition of the wood, the main components account for 97–98% (saccharide—cellulose 49%; hemicellulose 24%; aromatic—lignin 24%), and the accompanying components account for 2–3% (e.g., resin). This work focuses on the tree resin.

The results of the energy analysis are presented in terms of the combustion heat values of the resins from *Pinus sylvestris* L., *Picea abies* (L.) H. Karst, and *Larix decidua* Mill. (Figure 2), as well as the pulp, an intermediate in wood production, and turpentine, a component of the resin (Table 2). For the other components of tree biomass, the energy parameters are relatively well known [37,65]. The heat values of pine, spruce and larch are quite high, and tree resin is probably the substance with the highest energy content out of all tree components in plants. The mean combustion heat values of pine, spruce, and larch are, respectively, $38.591 \pm 1.307$ MJ·kg$^{-1}$, $38.373 \pm 1.521$ MJ·kg$^{-1}$, and $38.326 \pm 0.975$ MJ·kg$^{-1}$. The differences between the combustion heat values of pine, spruce, and larch are statistically insignificant ($p < 0.509$).

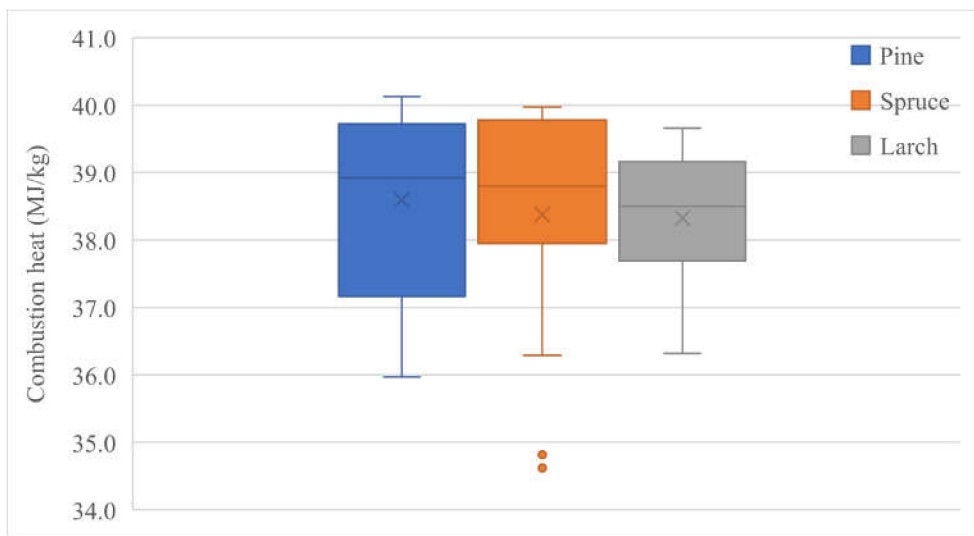

**Figure 2.** Resin combustion heat of pine, spruce, and larch. $n = 19$.

**Table 2.** Resin combustion heat of bleached and unbleached pulp and of turpentine.

| Component of Wood | Combustion Heat (MJ·kg$^{-1}$) | Ash (%) |
|---|---|---|
| Bleached pulp | $17.319 \pm 0.025$ | 1.29 |
| Unbleached pulp | $15.955 \pm 0.036$ | 8.21 |
| Turpentine | $39.773 \pm 0.027$ | 2.83 |

The resin energy values measured were further compared with cellulose associated with paper production, and can be supposed to be a by-product. The readings of bleached pulp (cellulose) produced by the Kraft sulfate process may serve as the energy reference standard for tree biomass. The mean combustion heat values of bleached ($17.319 \pm 0.025$ MJ·kg$^{-1}$) and unbleached pulp ($15.955 \pm 0.036$ MJ·kg$^{-1}$) (Table 2) are 2.2 and 2.4 times lower compared to the lowest value of the investigated tree resin (larch $38.326 \pm 0.975$ MJ·kg$^{-1}$).

The combustion heat of turpentine ($39.773 \pm 0.027$ MJ·kg$^{-1}$), a constituent produced as a by-product in the sulfate technological procedure, is 2.3 and 2.5 times higher than those of white and brown pulp. This figure is even slightly higher than that of tree resin, and this may be due to the different representations of the volatile components of turpentine (Table 3; Figure 3).

**Table 3.** Composition of turpentine.

| Constituent | Content (%) |
|---|---|
| Dimethylsulfide | 25.54 |
| Dimethyldisulfide | 0.14 |
| α-pinene | 36.28 |
| Camphene | 0.72 |
| β-pinene | 9.74 |
| Myrcene | 0.15 |
| δ-3 carene | 9.34 |
| P-cymene | 0.29 |
| D-limonene | 2.70 |
| α-terpinene | 0.93 |
| Γ-terpinene | 0.18 |
| Terpinolene | 0.94 |
| Unidentifiable | 3.95 |
| Unidentifiable (heavier than terpinolene) | 9.1 |

Method: Gas chromatograph—Agilent 7890A Yield and technical characteristics of the tapping operations.

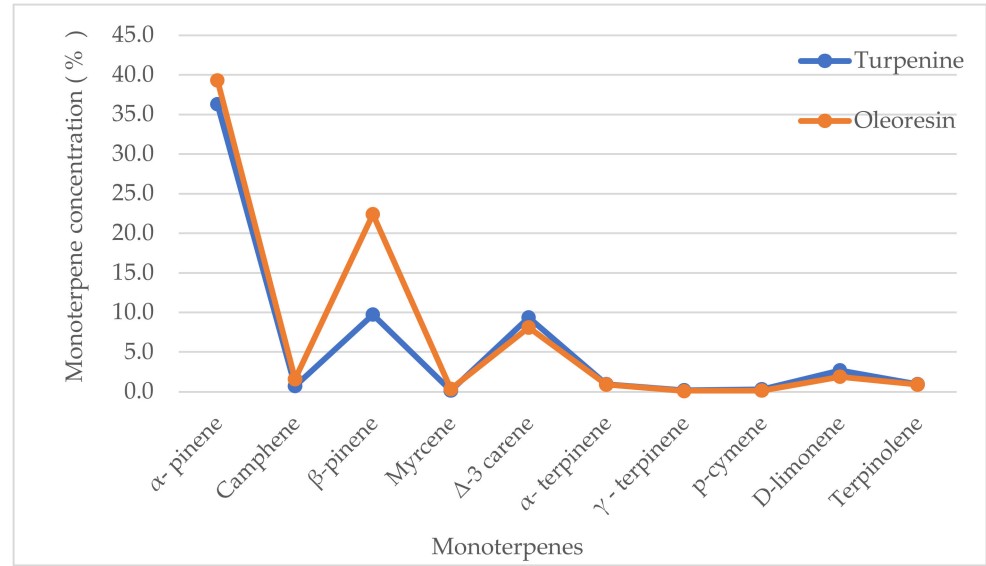

**Figure 3.** Monoterpene concentrations in turpentine and oleoresin of *Pinus sylvestris* L. (oleoresin values taken from [52]).

The properties of oleoresin are determined mainly by the proportion of volatile monoterpenes. The ratio of volatile to non-volatile components affects the physical and defensive properties of oleoresin. The volatile and more fluid components of resin enable the movement of the more viscous non-volatile components. The non-volatile di- and triterpenoid or phenolic components increase the viscosity and rate of crystallization of the

oleoresin. These properties affect the rate of resin flow and the ability of the resin to trap and immobilize enemies or coat wounds in tree trunks, which has been considered a first line of tree defense [10].

In this work, the analysis of the representations of turpentine components was performed, and the concentrations measured were compared to those in the oleoresin of *Pinus sylvestris* L. (Figure 3). The highest levels in turpentine were recorded for α- and β-pinenes (together making up 46% of the total content), but were still lower than that of oleoresin (62%). Δ-3-carene and D-limonene were in the range of 3–10%. Further resin constituents observed were below 1%, namely, terpinolene, camphene, α- and γ-terpinene, p-cymene, and myrcene. Although turpentine is prepared by a distillation process, its composition is rather similar to that of the oleoresin of *Pinus sylvestris* L. [52]. A similar composition of monoterpenes was found in needles and cortical oleoresin [66]. In needles, the major constituent was β-pinene, and it was α-pinene in the latter. In general, the content of monoterpenes in needle oleoresin decreases from winter to summer, while the concentration of sesquiterpenes increases. In cortical oleoresin, the case was the reverse. This descending trend of monoterpene concentrations in needles was most likely caused by their ascending release into the atmosphere, with an increasing temperature up to the summer period. The monoterpene content, which represents the main components of the natural emission load, has its own effect on the course of global warming.

Furthermore, the combustion value of resin was compared to that of the tree wood. The combustion heat of tree wood irrespective of tree species ranges from 18.000 to 20.000 MJ·kg$^{-1}$ [67]. Similar results were stated for softwood [38,68]. The lowest value was found in eastern white cedar branches (18.668 MJ·kg$^{-1}$) and the highest was found in black spruce treetop (21.562 MJ·kg$^{-1}$); the mean value of all softwood components is 20.178 MJ·kg$^{-1}$. For the hardwood, the span is wider—the lowest value is 17.230 MJ·kg$^{-1}$ in Manitoba maple foliage, and the highest one is 21.119 MJ·kg$^{-1}$ in White birch foliage; the mean of all components is 19.146 MJ·kg$^{-1}$. An even wider span of values was given [69] in a statistical summary of the calorific values of 402 species of wood of 246 genera, and 66 species of bark of 33 genera, based primarily on literature surveys. The calorific values range from 15.584 to 23.723 MJ·kg$^{-1}$ in hardwood and from 18.608 to 28.447 MJ·kg$^{-1}$ in softwood. Excluding the highest values in softwood, the resin combustion heat is almost twice as high as that of wood.

In the case of refined liquid hydrocarbon fuels such as petrol or diesel, the mean energy value of conifer resins is close to the lower end of the combustion heat range of diesel (about 41.900 MJ·kg$^{-1}$) and petrol (43.500 MJ·kg$^{-1}$). However, the maximum measured value of pine resin combustion heat (40.109 MJ·kg$^{-1}$) is only 5% lower than hydrocarbon fuel's combustion heat (1.8 MJ). High-calorific solid fuel such as coke registered only 70% of the heating value measured in the pine resin. Black coal achieved only 65%, and brown coal achieved only 36%.

## 4. Discussion

The use of renewable energy sources is becoming increasingly important to achieving the changes required to address the impacts of global warming. In the context of current European Union policy, woody biomass is expected to be an important energy resource in the near future. Up to now, tree biomass has been investigated mostly for the energy utilization of tree body components, such as the branches and stumps of broadleaves and conifers [37,68]. However, several studies have specifically focused on the enhancement of the fuel characteristics of woody biomass, such as wood density, volatile matter, the calorific value of wood [70], the fertilization of *Picea abies* stands [71], landscaping, and bioesthetic planning [72]. Further, the individual constituents of tree resin, such as rosin and turpentine, have been studied [73–75]. However, only a limited number of studies have focused on a macroergic material such as tree resin.

This paper reveals that the combustion heat values of pine, spruce, and larch are 38.591 MJ·kg$^{-1}$, 38.373 MJ·kg$^{-1}$, and 38.326 MJ·kg$^{-1}$, respectively, and the difference be-

tween figures is statistically insignificant ($p < 0.509$). The analysis of the energy components of the wood biomass shows that the combustion heat of oleoresin (Figure 2) is higher than that of bleached cellulose (2.2-fold) and unbleached cellulose (2.4-fold). Only the calorific value of turpentine is higher, by 4%, but the percentage of turpentine in spruce wood is only 0.1–0.2%. The resin content of the tree's wood is more than 1–2%.

Similar results, albeit showing large variation, were recorded by Howard [41] and Ivask [76]. While Howard [41] reported lower heating values in a range of 34.89–37.22 MJ·kg$^-$ in tree pine resin, Ivask [76] recorded higher values in spruce resin, namely, $40.10 \pm 0.62$ MJ·kg$^{-1}$. These differences in energy values in tree resin can be explained by differences in the ratio of cellulose and lignin [41], by topographical [77], ecological [78], seasonal [79], and environmental aspects, and naturally by the conditions and methods of determination. The variation in values [41] was large, as the samples were difficult to mix thoroughly, and were therefore not homogeneous. Ivask [76] predominantly investigated the influence of seasonal aspects, as well as the local and climatic conditions, on the energy parameters of tree wood components. Spruce tree resin combustion heat was determined in only six samples, and the author [76] himself admitted that a random estimate of calorific values yields very little information.

Although oleoresin is obtained by tapping in small quantities, it takes place over almost the entire year, and can be obtained from a living tree. The quantity and quality of the oleoresin are also determined by a tapping process. Resin sampling from pine trees has occurred since the mid-19th century [80]. However, it is necessary to mention that, since the 1980s, there has been a decline in oleoresin sampling due to lower purchase prices. This decrease in price was caused by the advent of cheaper competing commodities produced from crude oil. However, nowadays, oleoresins with high calorimetric values might return in economies that utilize green biofuels and bioproducts from non-food feedstocks [81]. There are four methods of oleoresin tapping; the American method is the most effective and provides 2.5 times the yield of the Chinese method (Table 4). The largest quantities of resin have been tapped from pine trees.

**Table 4.** Yield and technical characteristics of the tapping operations. The values are valid for pine trees.

| Location | Brazil | China |
|---|---|---|
| Tapping Technique | American | Chinese |
| Density (trees/ha) | 800 | 700 |
| DBH (cm) | 25 | 15 |
| Season (months) | 9 | 5.6 |
| Years in production | ~20 | 5 to 7 |
| Yield per time (g/day) | 19.7 | 11.2 |
| Trees tapped per worker | 7000 | 1500 |
| Hectares tapped per worker | 8.75 | 2.18 |
| Metric tonnes produced per worker | 35 | 3 |
| Pine resin (kg/year) | 5 | 2 |

Legend: Modified table [50].

The American sampling method provides 5 kg of resin per year under optimal conditions. This amount fluctuates according to local conditions; for example, in Portugal, this amount was reduced by almost half [82]. Some investigations have been carried out with the aim of assessing the economic viability of performing resin-tapping operations [21,33–35]. However, the highest increase in extractive contents in *Pinus elliottii* biomass was achieved by using a 2% paraquat-cation stimulant [83]. In the low 152 cm bolt, there was an 884% rise in resin acid amounts and a 2360% rise in turpentine values, and these values underwent 273% and 684% increases, respectively, in the whole stem. By applying a 2% paraquat-cation stimulant in the American tapping method, the yield might be as much as 18.65 kg of oleoresin per pine tree. Per hectare, this yields a total amount of

14,920 kg (575.78 GJ). Increasing resin production occurs with damage to the health status of the sample trees.

Tapping, irrespective of the method employed, causes intensive wounds in tree stems, leading to wood deformation. Tapped trees, compared to non-tapped ones, after one tapping, show a decrease in mean tree ring width by 14.1% (an average tree ring width of $2.41 \pm 0.85$ mm was reduced to $2.07 \pm 0.7$ mm), but this decrease is only by 6% in late wood [84]. Decreases in the volume of the wood are visible; however, this damage is negligible compared to the energy that is stored in the tapped resin.

Due to its diverse applications, the *Pinus* genus is considered one of the most important commercial timber species [85]. Today, it is well established that resin properties depend mostly on factors such as genetic background and environmental effects [42,55]. Its low technical requirements for planting [19] make *Pinus* one of the most suitable woody species for cultivating and restoring marginal areas, as well as abandoned and degraded agricultural lands [86].

In the 2015–2030 period, incremental abandonment is expected to reach 4.2 Mill. ha net (approximately 280,000 ha per year on average) of agricultural land, bringing the total abandoned land to 5.6 Mill. ha by 2030 (3% of total agricultural land). Arable land is projected to account for the largest share of abandoned land (4.0 Mill. ha; 70%), followed by pasture (1.2 Mill. ha; 20%) and permanent crops (0.4 Mill. ha; 7%). Nearly a quarter ($\approx$1.38 Mill. ha) of all agricultural areas in mountainous areas in the EU will probably be abandoned [87]. This supposes that half of the abandoned agricultural land in mountainous areas (700,000 ha) will be afforested, and pine oleoresin will be tapped from an adult pine stand growing on this land. The pine oleoresin collected by the American method using a 2% paraquat-cation stimulant over one year will yield an energy value of 403.044 PJ, which would provide 0.82% of the fossil coal energy and 0.24% of the total energy (160–180 EJ) required worldwide (as of 2018). This energy is produced by more than 15.5 Mt of coal, with each metric tonne of coal producing 1700–1800 $m^3$ of $CO_2$, thus exacerbating the problem of global warming [88]. If this coal is replaced by tapped resin via the combustion process, the atmospheric load will be reduced by 27.1 $Gm^3$ $CO_2$, i.e., 53.7 Mt, as its origin is not fossil fuel (Table 5).

**Table 5.** Resin and carbon dioxide quantities associated with a 700,000-ha pine forest stand.

| Area | Resin Tapped Over One Year | | C Sequestered by Pine Trees | CO₂ Released from Coal (Equivalent of Resin Amount) | |
|---|---|---|---|---|---|
| (ha) | (Mt) | Q (PJ) | (Mt) | (Gm³) | (Mt) |
| 700,000 | 10.444 | 403.044 | 1.179 | 27.128 | 53.700 |

Legend: 4.320 Mt $CO_2$ is equivalent to 1.179 Mt C; $\rho_{CO_2} = 1.98$ kg $m^{-3}$; $m_{coal} = \left( \frac{Q_{resin}}{Q_{coal}} \right) * 10.444 = 15.5$ (Mt).

Further, forests have also been recognized for the other services they offer, such as the ability to store carbon and mitigate the impacts of climate change [89]. Forest plantations are beneficial in mitigating climate change and reducing airborne emissions only through stringent management. A 700,000 ha *Pinus sylvestris* plantation with a net ecosystem exchange (NEE) rate of 1.684 t C $ha^{-1}$ $yr^{-1}$ sequesters 1.179 Mt of carbon per year (4.320 Mt $CO_2$). A NEE rate of 1.684 t C $ha^{-1}$ $yr^{-1}$ converges with the average annual increase in C in a Scots pine stand equaling 1.234 t $yr^{-1}$ (105.42 t/85.46 years) [90]).

Although the aforementioned example of afforestation is only theoretical, it reveals the possibilities of oleoresin utilization in the environmental field. All constituents of biomass are photosynthesized in plant leaves from $CO_2$, water, and absorbed solar energy. A specific feature of biomass is that its combustion produces the same amount of the greenhouse gas that was absorbed during photosynthesis. Biomass is neutral in terms of $CO_2$ emissions. The establishment and management of forests as a source of oleoresin also supports the removal and storage of $CO_2$ from the atmosphere, and offsets the increase in the anthropogenic emissions of greenhouse gasses (GHGs), consequently reducing the rate of global warming and mitigating the impacts of climate change [52,89]. A plantation system is a good means of climate change mitigation [91,92].

Generally, according to the essence of the production processes, feedstocks are classified as first-, second-, or third-generation. Oleoresin, by its lignocellulosic nature, can be assigned to the second generation of feedstocks. Third-generation feedstocks are produced from algae, sewage sludge, and municipal solid waste (this is omitted) [93]. The prices of these feedstock generations differ. First-generation feedstock supply is supposed to be available at production costs of EUR 5–15 GJ$^{-1}$ compared to EUR 1.5–4.5 GJ$^{-1}$ for second-generation feedstocks [94]. The production cost of first-generation feedstocks is approximately triple that of the second-generation ones. The difference is slightly smaller between the cost of residues in agriculture and forestry (EUR 1–7 GJ$^{-1}$ and EUR 2–4 GJ$^{-1}$, respectively).

For the 2008–2011 period, the production cost of oleoresin was nearly double that of lignocellulosic crops, i.e., USD 0.2–0.4 kg$^{-1}$ ≡ EUR 4.6–9.2 GJ$^{-1}$ [95]. However, when oleoresin tapping was carried out by the American tapping method with an application of a 2% paraquat-cationic stimulant, the production cost was significantly reduced. If we take into account the ratio between the energy stored in oleoresin tapped from 1 ha of pine stand (575,78 GJ yr$^{-1}$) and that stored in wood from 1 ha of SRC of willow (172 GJ yr$^{-1}$), the oleoresin production per unit amount is the cheapest, at EUR 1.3–2.75 GJ$^{-1}$ (Table 6). Further investigations are needed to demonstrate this.

**Table 6.** Comparison of energy values of renewable energy sources.

| Species | Yield | Heating Value | Energy per Hectare | | Production Cost |
|---|---|---|---|---|---|
| | (t) | (MJ kg$^{-1}$) | (GJ) | (MWh) | (€/GJ) |
| Feeding [a] sorrel | 10.0 | 16.00 | 160.00 | 44.44 | |
| Common [b] reed | 12.7 | 17.45 | 221.60 | 61.56 | 1.5–4.5 |
| Miscanthus [c] | 14.0 | 17.60 | 246.40 | 68.45 | |
| Pine resin [d] | 14.9 | 38.59 | 575.80 | 159.94 | 1.3–2.75 |

Legend: [a] a hybrid of *Rumex patientia* L. (maternal line) and *Rumex tianschanicu* (paternal line) species [96]; [b] *Phragmites australis* (Cav.) Trin. [97]; [c] *Miscanthus gigantheus* [98]; [d] *Pinus sylvestris*.

Biofuels are a potentially low-carbon energy source, but whether biofuels offer carbon savings depends on how they are processed [92]. Tree resin has been proven to be an excellent renewable energy source that has not yet been used for this purpose. Compared to previously investigated energy sources, resin has the highest energy content per hectare (Table 6). The production of oleoresin is not the primary goal, but the volume of the greenhouse gas $CO_2$ will be reduced with afforestation; consequently, the rate of global warming will be dampened. Afforestation associated with the conversion of marginal agricultural or forestry lands to purpose-grown crops is an important practice used to lower the rise in atmospheric $CO_2$ concentration, due to a forest's ability to fix carbon in tree biomass and in soil [52,99,100].

While wood reduction in forests requires the removal of woody biomass, utilizing it for power generation reduces overall emissions by 98% in comparison with slash pile burning [101]. Energy production in the forest is more environmentally friendly. For example, regrowth energy in a forest stand generated from wood leads to a production of 0.057 metric tonnes of $CO_{2e}$ per MWh, compared with the average US rate of 0.60 metric tonnes of $CO_{2e}$ per MWh [102]. Striking relations also exist between the annual amounts of carbon sequestered by a pine stand, the resin tapped from pine trees by the American method, and the carbon dioxide released from coal with the same energy content as tapped resin (Figure 4).

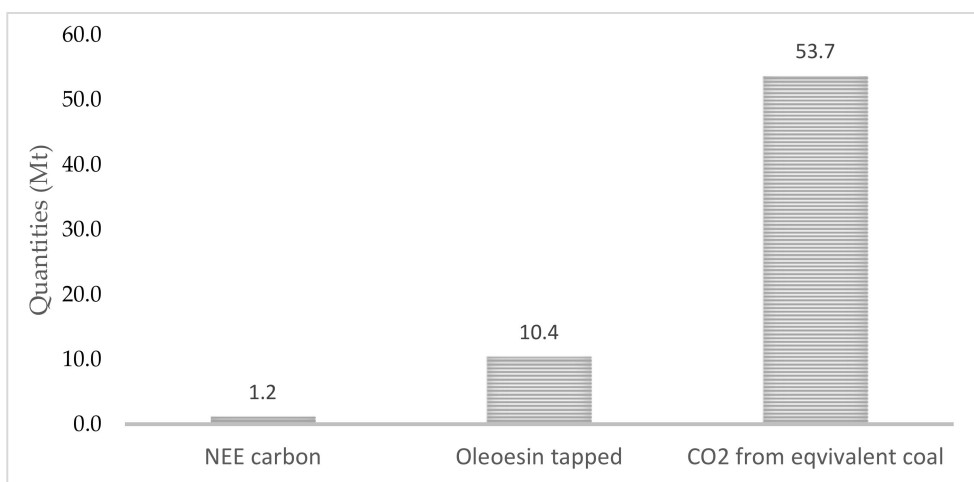

**Figure 4.** Quantities of net ecosystem exchange (NEE) C accumulated in stand, oleoresin obtained from a pine trees stand and $CO_2$ from coal (energy equivalent to resin tapped over one year; 700,000 ha).

The lowest amount was recorded for NEE carbon sequestered by a tree stand. The extracted resin and the released $CO_2$ held, respectively, 9 times and 45 times the quantity of the carbon sequestered. This means that 1 tonne of oleoresin corresponds to approximately 5 tonnes of $CO_2$. If the oleoresin is burned as a renewable energy source instead of coal, the emission load in the atmosphere would be reduced by roughly 5 tonnes of anthropogenic $CO_2$. (This is valid provided that $CO_2$ generated from oleoresin is environmentally neutral)

Tree resin production is directly related to bioeconomic issues. The bioeconomy may be conceived as a prime way of engaging with ecological modernization, i.e., economic and technological modernization that seeks to address perceived environmental issues [103–105]. The bioeconomy may well be an inevitable transition if fossil resources are to be phased away [106]. The bioeconomy is attracting interest as a conceivable win–win solution for green growth. The European Bioeconomy Strategy supports the production of renewable biological resources and their conversion into vital products and bioenergy in order to satisfy the 2030 Agenda and its Sustainable Development Goals [107,108]. It represents a wide range of opportunities for sustainable development in bio-based industries [109,110], which encompass various sectors, including agriculture and forestry.

The bioeconomy aims to substitute fossil resources with bio-based alternatives, such as tree resin applications for energy. This energy utilization can be evaluated by the socio-economic indicator of the bioeconomy (SEIB) [111]. The circular economy (CE) concept aims to reduce resource use and consumption, favoring reuse and recycling activities and aiming to minimize waste and emissions (environmental risks). To make progress toward the CE, it is essential to prepare accurate estimates of the environmental/economic and ethical dimensions of proposals to support this transition. So far, among the potential energy forms to be derived from biomaterials, biogas is of the greatest significance due to its ability to transform organic feedstocks into biomethane ($CH_4$).

The bio-$CH_4$ potential of several European cities was estimated, and a large share of this potential can be used as vehicle fuel and can therefore help the European Union (EU) to achieve its Paris Agreement commitments within the transport sector [112,113]. Currently, the transport sector is responsible for a third of the global energy demand, and one-sixth of global GHG emissions. [112,114]. This sector is currently dominated by the usage of fossil fuels in Europe. The experiment with bio-$CH_4$ yielded the following.

The production of bio-$CH_4$ in Portugal would only be profitable under potential government incentives in the form of feed-in premia above 15.42 EUR.GJ$^{-1}$ (55.5 EUR.MWh$^{-1}$) [115]. Selling $CO_2$ for a price of 46 EUR.t$^{-1}$ $CO_2$ would also be profitable. In 2017, a total of $1.94 \times 10^9$ (American billion) cubic meters of bio-$CH_4$ was produced in Europe. In 2050, the European biomethane potential will be $95 \times 10^9$ cubic me-

ters [116,117]. This bio-$CH_4$ production, showing a trend of 50-fold enlargement from 2017 to 2050, indicates the possible energetic use of tree resin due to its price. The bio-$CH_4$ production cost is approximately five times the cost of tree resin production (1.3–2.75 EUR.$GJ^{-1}$). The application of tree resin for energy utilization is associated with environmental issues, and may have social utility in such cases as in southern Brazil [110,118–120].

The objective of the 2015 Paris Climate Agreement is to hold average global warming to well below 2 °C above preindustrial levels. While this objective is formulated at the global level, the success of the agreement depends on the implementation of climate policies at the national level. Each country submits nationally determined contributions (NDCs) [121]. In the EU, in order to achieve conditional NDCs, greater reductions are necessary than those achieved by national policies only. The implementation of conditional NDCs (NDC scenario) is projected to result in 51.9 (50.4–57.4) $GtCO_{2eq}$ GHG emissions by 2030 [121]. When tree resin tapped from 700,000 ha over one year (the previous example) was used for heat purposes, 0.1% of the 51.9 Gt $CO_2$ would be reduced. For a period of 15 years, it would be 1.55%. The same is valid for the year 2050. Several means to achieve an 80% reduction in GHG emissions, implying an 85% decline in energy-related $CO_2$ emissions (including those from transport), have been examined (Table 7). This is where the oleoresin application seems appropriate—it relates to reduced GHG emissions and to improved energy efficiency.

**Table 7.** Plans to reduce greenhouse gas emissions into the atmosphere, to increase the use of renewable energy sources and to improve energy efficiency by 2050.

| Parameters Examined | 2020 Climate and Energy Package [122,123] | 2030 Climate and Energy Framework [124] | Energy Road 2050 [125] |
|---|---|---|---|
| Cut in GHG from 1990 levels | 20% | 40% | 80% |
| EU energy from renewables | 20% | 32% | 66% |
| Improvement in energy efficiency | 20% | 32.5 | 75% |

The use of tree resin for energy is only possible through stringent bioeconomic management. Its benefit is to reduce the additional load of $CO_2$ into the atmosphere, as aforementioned. On the one hand, oleoresin combustion is associated with the release of neutral $CO_2$, and on the other hand, the abandoned and degraded land is used for the afforestation of new forest stands that accumulate $CO_2$ for at least one rotation. However, the technological processing of the resin needs to be developed.

## 5. Conclusions

This paper focuses on the energy value (calorific or heating value) of certain components of the tree biomass of pine, spruce, and larch. The energy parameters of the trees examined were determined by calorimetry. The components of wood pulp (bleached and unbleached) and turpentine were obtained from the pulp and paper processing plant Mondi SCP (Slovak Cellulose Paper mill), Ltd. in Ruzomberok. Samples of tree resin were taken from V-shaped wounds carved in the bark of trees growing on the Cernova research stand nearby Ruzomberok. The mean combustion heat values of the tree resins (pine: 38.591 ± 1.307; spruce: 38.373 ± 1521; larch: 38.326 ± 0975 MJ·$kg^{-1}$) are not statistically different from each other. However, these values are double the heating value of forest tree wood and its components.

The combustion heats of bleached pulp (cellulose) produced by the Kraft sulfate process are supposed to be the reference standard for tree biomass. The average combustion heats of bleached (17.319 ± 0.025 MJ·$kg^{-1}$) and unbleached (15.955 ± 0.036 MJ·$kg^{-1}$) pulps are 2.2 and 2.4 times lower than those of the investigated tree resin samples. The highest energy value was recorded for turpentine (39.773 ± 0.027 MJ·$kg^{-1}$), but its technical processing is considerably more complicated than that of the resin.

The quality and quantity of oleoresin was influenced by tapping. The American method was best, providing 5 kg of oleoresin per tree over one year. However, by using the paraquat-cation stimulant, this amount was enhanced to 18.65 kg of oleoresin per tree over a year; thus, a 1 ha pine stand provides 14,920 kg.

Tree resin was shown to be an excellent renewable energy source that has not yet been used for this purpose. Compared to the previously investigated energy sources (feeding sorrel: 160 GJ ha$^{-1}$; common reed: 221.6 GJ ha$^{-1}$; Miscanthus *gigantheus*: 246.4 GJ ha$^{-1}$), pine resin, with 575.8 GJ ha$^{-1}$, showed the highest energy content per hectare.

The production price of oleoresin was compared to that of other feedstocks. The production cost of first-generation feedstocks is triple that of second-generation ones (EUR 5–15 GJ$^{-1}$ compared to EUR 1.5–4.5 GJ$^{-1}$). When using a 2% paraquat-cation stimulant, the production cost of oleoresin is the lowest (EUR 1.3–2.75).

The potential contribution of forest expansion to the sequestration of carbon and the utilization of renewable forest resources is well known. The tapping of resin and the expansion of its industrial uses can further enhance the carbon sequestration potential of forest resources and expand the forest ecosystem services. An average NEE rate of 1.684 t C ha$^{-1}$ yr$^{-1}$ was calculated at three geographic locations (lat. from 42° N to 62° N). If one tonne of oleoresin was burned instead of the energy equivalent amount of coal, the atmosphere would be spared more than 5 Mt $CO_2$.

The use of oleoresin for energy is only possible through stringent bioeconomic management. The circular economy aims to reduce resource use and consumption to minimize emissions. So far, among the potential energy forms derived from biomaterials, biogas is of the greatest significance. However, its production is only profitable under government incentives (15.42 ERU.GJ$^{-1}$), which is about five times the cost of tree resin production. The application of oleoresin for energy use is also associated with environmental security, and has a social role. It can also be helpful to reduce $CO_2$ GHG emissions and improve energy efficiency.

Tree oleoresin, as a macroergic substance, approaches the heating parameters of liquid hydrocarbon fuels such as oil and petroleum products. Therefore, oleoresin can play an important role in the supersession of fossil fuels. Resin quantity and quality might be controlled by selecting suitable tree taxa of high oleoresin content, or its energy content in a tree can be increased by applying a suitable stimulant. The aim of this work is the cultivation of energy-modified biomass, a revolutionary shift in the production of renewable energy sources.

**Author Contributions:** The basic idea and the overall scope of this research was under the responsibility of J.D. Chemical processing of biological material, chemical analyses and writing the article managed J.M. All authors have read and agreed to the published version of the manuscript.

**Funding:** This research was funded by the 2/0013/17 VEGA grant—"Ecosystem services to support landscape protection in conditions of global change" in the period 1.1.2017-31.12.2020.

**Data Availability Statement:** In this manuscript, all the necessary data were published, except for individual combustion heat values of studied trees, where average values were depicted in a graph.

**Acknowledgments:** The authors have reviewed and agree with the contents of the manuscript and there is no financial claim against publication. It is confirmed that the paper is an original work and is not subject to review in any other publication.

**Conflicts of Interest:** The authors have declared no conflict of interest.

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
