# Peer review of "Tree Resin, a Macroergic Source of Energy, a Possible Tool to Lower the Rise in Atmospheric CO2 Levels"

_sustainability, doi:10.3390/su14063506_

Round 1

Reviewer 1 Report

Review of article (manuscript) for the journal SUSTAINABILITY
Manuscript Number: sustainability-1347612
Manuscript submitted: 11 August 2021 (19 p.)
Title of the manuscript (Original Article):
Tree Resin as the Most Macroergic Constituent of Conifers – a possible means to mitigate global warming and climate change   
Authors:
Demko, J. and Machava, J.
_________________________________________________________________
The manuscript gives a wide view on tree resins and their utilization mainly as the ecologically suitable energetic sources having high heating values. The manuscript is well written – evidently better as its first version for the journal Forests MDPI from 2020 – so I can recommend this version for publication in your scientific journal.

14 August 2021                                                                           Reviewer

Author Response

The manuscript gives a wide view on tree resins and their utilization mainly as the ecologically suitable energetic sources having high heating values. The manuscript is well written – evidently better as its first version for the journal Forests MDPI from 2020 – so I can recommend this version for publication in your scientific journal.

Thanks for the positive evaluation of our manuscript, Jan Machava.

Reviewer 2 Report

From the abstract: "The main goal of this work is to determine the energy content of the resin of spruce, pine and larch and 10 wood components – pulp and turpentine."

And yet the paper is 19 pages, with 144 references. With rambling discussion of climate change mitigation, production costs of resin, components of resin, etc.

A succinct, technical note on the calorific value of resin might be worth publishing. 

Author Response

The article was revised several times based on the recommendations of previous opponents. This relates to prove that the resin is a renewable energy source. Especially, to confirm that its production value is comparable to other energy sources (The first- and second-generation feedstocks). When applying a stimulant, the cost of the resin production value is the lowest (this is a part of questions requested by previous opponents). We state the benefits of the article:

The contribution of the article is in the field of an afforestation. The deforestation problem leads to increasing the global atmosphere temperature, that may lead to the climatic change. Connecting the afforestation process with the tapping of tree resin for energy purposes is further benefit of resin application.

The tapping of tree resins did not only lead to a sustainable renewable energy source, but also led to lower the terpenes content in VOC emissions.

Extensive discussion as well as the work conclusion were drawn up in the frame of recommendations of previous Reviewers from 2020. With the article we have reached the situation in which the requests of the 2. Reviewer are not in line with previous opponents. It is directly unsolvable state for us. Either rejected will be the comments of previous opponents or the comments of the current 2. Reviewer.

For this reason, we have expanded the objectives of this manuscript to be consistent with the contents of the discussion and conclusion (there was a lack of the manuscript - the objective of the manuscript was not in line with the contents of the discussion. The discussion seemed to be excessive.).

The goal of this article in the last paragraph of Introduction:

The goal of this work is to investigate of oleoresin of studied trees as a sustainable renewable energy source, which might be very useful at lowering global warming and climatic change damping. Therefore, the studied chemical composition of oleoresin and the factors influencing the values of oleoresin combustion heat will be assessed in the frame of energy issues solution – afforestation of abandon areas, comparation of a production value of other feedstocks, the reduction of VOC in the atmosphere, etc. Combustion heat of tree resin will also be compared to other components of tree wood.

Thus, the use of tree resin for energy purposes leads to the mentioned contributions that were also reflected in the Environmental Field. For this reason, the assets of this comprehensive article are greater than that of the Technical Report.

After this correction, we want to ask the 2. Reviewer to review his attitude to the publication of this article. Thank you very much for your helpfulness.

                                                                                    Jan Machava

Reviewer 3 Report

I find that is this is a very interesting article that deals with a topical issue of turning the whole energy mentallity to green solutions.

Author Response

Thanks for the positive evaluation of our manuscript, Jan Machava.

This manuscript is a resubmission of an earlier submission. The following is a list of the peer review reports and author responses from that submission.

Round 1

Reviewer 1 Report

I still think the data are appropriate for a technical note. The wider discussion doesn't fit well in the current paper - the topic may be worthy of wider discussion but it should be a separate paper.

Specific comments:

L128+ Not impossible at all – you refer to Howard, eg. Also to Chen’s paper, which also has references within. Also, Kossuth, Susan V., Donald R. Roberts, Jacob B. Huffman, and Shih-Chi Wang. "Resin acid, turpentine, and caloric content of paraquat-treated slash pine." Canadian Journal of Forest Research 12, no. 3 (1982): 489-492.

Figure 2 would be better as box and whisker plot or similar.

Reviewer 2 Report

The manuscript gives a wide view on tree resins and their utilization mainly as the ecologically suitable energetic sources having high heating values. Manuscript is well written. More comments related to the first version were implemented.

13 February 2020                                                                             Reviewer